# Dipeptide Extract Modulates the Oxi-Antioxidant Response to Intense Physical Exercise

**DOI:** 10.3390/nu14122402

**Published:** 2022-06-09

**Authors:** Agnieszka Zembron-Lacny, Edyta Wawrzyniak-Gramacka, Anna Książek, Aleksandra Zagrodna, Wiesław Kopeć, Małgorzata Słowińska-Lisowska

**Affiliations:** 1Department of Applied and Clinical Physiology, Collegium Medicum University of Zielona Gora, 65-046 Zielona Gora, Poland; e.gramacka@cm.uz.zgora.pl; 2Department of Biological and Medical Basis of Sport, Faculty of Physical Education and Sport, Wrocław University of Sport and Health Sciences, 51-612 Wrocław, Poland; anna.ksiazek@awf.wroc.pl (A.K.); aleksandra.zagrodna@awf.wroc.pl (A.Z.); malgorzata.slowinska-lisowska@awf.wroc.pl (M.S.-L.); 3Department of Animal Products Technology and Quality Management, Wrocław University of Environmental and Life Sciences, 50-375 Wrocław, Poland; wieslaw.kopec@upwr.edu.pl

**Keywords:** inflammation, glutathione, hydrogen peroxide, nitric oxide, oxidative stress

## Abstract

Exposure to intense physical exercise increases reactive oxygen and nitrogen species production. The process can be modulated by dipeptide bioavailability with antioxidant scavenger properties. The effects of dipeptide intake in combination with physical exercise on the oxi-antioxidant response were examined in a randomized and placebo-controlled trial. Blood samples were collected from 20 males aged 21.2 ± 1.8 years before and after 14-day intake of chicken breast extract (4 g/day), which is a good source of bioactive dipeptides. A significant increase in the NO/H_2_O_2_ ratio was observed in the 1st and 30th minute after intense incremental exercise in dipeptides compared to the placebo group. Total antioxidant and thiol redox status were significantly higher in the dipeptide group both before and after exercise; *η*^2^ ≥ 0.64 showed a large effect of dipeptides on antioxidant and glutathione status. The level of 8-isoprostanes, markers of oxidative damage, did not change under the influence of dipeptides. By contrast, reduced C-reactive protein levels were found during the post-exercise period in the dipeptide group, which indicates the anti-inflammatory properties of dipeptides. High pre-exercise dipeptide intake enhances antioxidant status and thus reduces the oxi-inflammatory response to intense exercise. Therefore, the application of dipeptides seems to have favourable potential for modulating oxidative stress and inflammation in physically active individuals following a strenuous exercise schedule.

## 1. Introduction

Dipeptides, including carnosine and its methylated analogue anserine, are effective antioxidants which are able to influence reactive oxygen and nitrogen species (RONS) generation and the injury–repair–regeneration cascade in skeletal muscle [1,2]. Nitric oxide (NO) and hydrogen peroxide (H_2_O_2_) affect metabolic pathways including apoptosis, mitochondrial respiration and biogenesis, and—what has more recently been demonstrated—muscle stem cell proliferation [3]. However, NO and H_2_O_2_ play contradictory roles in tissue regeneration and repair, i.e., they lead to the recovery of tissue function, whereas the local persistence of NO and H_2_O_2_ sustained by infiltrated neutrophils may cause further oxidative injury to differentiating myoblasts and myotubes, thereby delaying complete health restoration [4]. To prevent or delay oxidative damage, several endogenous antioxidant systems are located in muscle tissue. These include α-tocopherol, histidine-containing dipeptides, glutathione and antioxidant enzymes such as glutathione peroxidase, catalase and superoxide dismutase. Dipeptides such as carnosine have been shown to be actively synthetised by muscle cells, and not merely deposited in skeletal muscles [5,6]. An increasing dietary intake of dipeptides enhances concentrations of carnosine and its methylated analogues in skeletal muscles. Several studies have shown that hydrolysates made up mostly of di- and tripeptides are absorbed more rapidly than amino acids and hydrolysates containing larger peptides and much more rapidly than whole foods. The nutritive value of a mixture of small peptides is greater than an amino acids mixture of a comparable composition [7]. Dietary dipeptides are absorbed by the enterocytes via the peptide transporter family (PEPT), then hydrolysed intercellularly and released as free amino acids into the circulation [8]. Dipeptides and dietary free amino acids stimulate the transcription of the PEPT gene and increase PEPT mRNA as well as protein abundance, leading to enhanced rates of peptide absorption [9].

Studies report that chicken breast extract (CBE) is a potential dietary source of dipeptides [10]. Its use has been shown to increase muscle carnosine content [11] and to improve high-intensity endurance performance through the attenuation of muscle fatigue and also the enhancement of post-exercise muscle regeneration [12]. The ergogenic potential of the pre-exercise intake of chicken extracts (containing both carnosine and anserine) has already been tested [11,13]. Suzuki et al. [11] showed that CBE, containing 0.4 g carnosine and 1.1 g anserine, enhanced the power output of the latter of two sets of repeated sprints. Blancquaert et al. [13] reported that markers of maximal exercise and muscle power during the initial stage of the Wingate test were significantly improved by pre-exercise 20–25 mg/kg of anserine and carnosine supplementation. So far, little is known on dipeptide intake in combination with intense physical exercise or on their influence on the oxi-inflammatory response [14,15,16,17]. Therefore, the present study was designed to evaluate the antioxidant and anti-inflammatory potential of the pre-exercise ingestion of chicken breast extract as a novel nutritional strategy in physically active individuals.

## 2. Materials and Methods

### 2.1. Participants

Thirty males were recruited for the study. Each participant underwent a thorough screening, including a complete medical evaluation. The exclusion criteria were as follows: serious orthopaedic injury (*n* = 1), nutrition supplements or medications (*n* = 3), low haemoglobin <12 g/dL (*n* = 2) identified at any point of the entire observation or the participants’ withdrawal from the project (*n* = 4). Eventually, twenty males aged 21.2 ± 1.8 years were included in the study (Table 1). Prior to the study, the males were randomly assigned in a double-blind manner to a comparison group (placebo) and an experimental group (dipeptides) based on a single sequence of random assignments using a shuffled deck of cards (placebo or dipeptide) with numbers from 1 to 20. A two-week washout period was introduced to avoid any possible interference of other nutritional supplements in the measured biochemical markers. Throughout the study, all subjects lived in the same accommodation and followed the same daily schedule (physical activity, study and meal times), sleeping time (seven hours daily). All the subjects were informed of the aim of the study and signed a written consent to participate in the project. The study was approved by the Bioethics Committee (No. 18/12/2008) at the University School of Physical Education Wrocław, in accordance with the Declaration of Helsinki.

### 2.2. Diet Analysis

The diet was evaluated using a food record method (foods and beverages consumed over 24 h for 7 consecutive days). The subjects were instructed on how to fill in the food record. The participants self-reported all the consumed foods and fluids and provided information about the times, types, and portion size of the meals and beverages they had consumed. The quantities of all the food/beverage items were reported in household measures or, if available, according to packaging details. The energetic value of food intake, as well as the ingredients of products and meals, were estimated by means of the Dieta 6.0 software (IZZ, Warsaw, Poland). Dieta 6.0 is a valuable application that calculates the intake of different nutrients, with bioavailability taken into account. To evaluate compliance with the recommended dietary intake, the supply of different nutrients was categorized as intake in accordance with the nutritional standards for the Polish population [18].

### 2.3. Chicken Breast Extract Intake

The chicken breast extract (CBE, 4 g per day for 14 days) or placebo (methylcellulose 4 g per day for 14 days) was consumed 1 h before the morning meal (Figure 1). CBE containing 40% of carnosine-related compound components (2:1 ratio of anserine to carnosine) was prepared following breast meat extraction with water (cold and at 80 °C) and spray drying. After 14 days of placebo or dipeptides intake, an incremental and progressive exercise test was performed to disturb the oxi-antioxidant equilibrium.

### 2.4. Body Composition

Body mass (BM), fat mass (FM) and fat-free mass (FFM) were estimated using the Tanita Body Composition Analyser BC-418 (Tanita, Tokyo, Japan) calibrated prior to each test session in accordance with the manufacturer’s guidelines. Duplicate measures were taken with the participant in a standing position; the average value was used for the final analysis. The recurrence of measurement amounted to 98%. The measurements of body composition were taken between 7.00 and 8.00 a.m. before blood sampling. The subjects were fasting during the body composition analysis.

### 2.5. Incremental Exercise Test and Maximal Oxygen Uptake

The maximal oxygen uptake (VO_2_max) was determined on a treadmill Trackmaster TM310 (Full Vision Inc., Newton, KS, USA) at a temperature of 22 °C and a relative air humidity of 60%. Breath-by-breath oxygen uptake was continuously recorded using the Oxycon Mobile ergospirometric system (Viasys Healthcare Inc., Conshohocken, PA, USA). The heart rate was continuously recorded during the test using a portable heart rate telemetry device: Polar Sport Tester T61 (Polar Electro, Helsinki, Finland). The test was incremental and progressive; all subjects commenced the test at a 4.5 mph running speed, which was increased by 0.5 mph every 2 min until the maximal level of recorded parameters was achieved.

### 2.6. Blood Sampling

Samples of the blood for laboratory tests were collected six times: before (initial analysis) and after 14 days of placebo or dipeptides intake (before exercise test and during post-exercise period including the 1st min, the 30th min, the 24th h and the 48th h), using S-Monovette tubes (Sarsted AG and Co. KG, Numbrecht, Germany). Within 20 min, they were centrifuged at 3000× *g* and +8 °C for 10 min. Aliquots of serum were stored at −80 °C. All samples were analysed in duplicate in a single assay to avoid inter-assay variability. The intra-assay coefficients of variation (CV) for the kits used were <8%.

### 2.7. Haematological Variables

Haematological parameters including red blood cell count (RBC), total white blood cell count (WBC), platelet count (PLT) and haemoglobin (HB) were determined using 3 diff BM HEM3 Biomaxima (Lublin, Poland).

### 2.8. Skeletal Muscle Damage and Lactate

Serum myoglobin (Mb) level was used as a marker of sarcolemma disruption and was evaluated using a Randox Laboratories kit (Crumlin, UK). The Mb detection limit was 5 ng/mL. Blood lactate was measured using the DP 310 Vario II mobile spectrophotometer Diaglobal (Berlin, Germany).

### 2.9. Oxi-Antioxidant and Inflammatory Variables

Nitric oxide (NO) and hydrogen peroxide (H_2_O_2_) were measured with Randox Laboratories kits (Crumlin, United Kingdom). NO and H_2_O_2_ detection limits were estimated at 0.5 µmol/L and 6.25 µmol/L, respectively. F2-izoprostan 8-epiPGF2a (8-isoprostane), as a marker of oxidative damage, was determined using a Cayman EIA kit (Ann Arbor, MI, USA) with a detection limit of 2.7 pg/mL. Total antioxidant status (TAS) was determined using a Randox Laboratories kit with detection limits at 0.125 mmol/L. Total glutathione (GSH_t_) and oxidized glutathione (GSSG) were measured using Randox Laboratories kits. GSH_t_ and GSSG concentrations were calculated using reduced glutathione as a standard. The detection limits for the GSH_t_ and GSSG were 0.1 μmol/L and 0.02 μmol/L, respectively. Thiol redox status was calculated according to the following equation: (GSH_t_–2GSSG)/GSSG. Before the measurement of glutathione, the blood samples were protected from oxidation according to the protocol of the Randox Laboratories method. C-reactive protein (CRP) concentration was determined using a commercial kit from DRG International (Springfield Township, Cinninnati, OH, USA) with a detection limit of 0.001 mg/L.

### 2.10. Statistical Analysis

Statistical analyses were performed using RStudio, version 4.1.2 [19]. The assumptions for the use of non-parametric or parametric tests were checked using the Levene and the Shapiro–Wilk tests to evaluate the homogeneity of variances and the normality of the distributions, respectively. The comparisons of repeated measurements inside groups were assessed using the t-Student test or the Wilcoxon Signed Rank Test depending on the compliance with the normality assumption. The significant differences in mean values between the groups (placebo and dipeptides) were assessed using one-way ANOVA. If the homogeneity and normality assumptions were violated, the Kruskal–Wallis non-parametric test was used. Additionally, eta-squared (*η*^2^) was used as a measure of effect size, which is indicated as having no effect if 0 ≤ *η*^2^ < 0.05, a small effect if 0.05 ≤ *η*^2^ < 0.26, a moderate effect if 0.026 ≤ *η*^2^ <0.64 and a large effect if *η*^2^ ≥ 0.64 according to our previous study [20]. The ideal sample size for a 30-person group was *n* = 28, whereas for a 20-person it was set at *n* = 20 to demonstrate significant differences (confidence level 95% and margin error 5%). Statistical significance was assumed at the level of *p* < 0.05.

## 3. Results

### 3.1. Diet Analysis

According to the standard of nutrition for the Polish adult population [18], our results did not demonstrate differences in intake of the major ingredients that might influence the oxi-antioxidant response. The mean daily consumption of energy, protein, carbohydrates and fat was 32.5 ± 6.5 kcal/kg body mass/day, 1.5 ± 0.3 g/kg body mass/day, 3.7 ± 0.8 g/kg body mass/day, 1.1 ± 0.3 g/kg body weight, respectively.

### 3.2. Haematological Variables

The results are presented in Table 2. The haematological variables fell within the referential range in all participants. No differences in RBC count, HB concentration and other parameters derived from peripheral blood analysis were observed between placebo and dipeptides groups.

### 3.3. Skeletal Muscle Damage and Lactate

The concentration of myoglobin (Mb) increased immediately after the exercise test from 132 ± 28 to 224 ± 59 ng/mL in the placebo group (*p* = 0.0028), and from 123 ± 41 to 273 ± 94 ng/mL in the dipeptide group (*p* = 0.003); Mb did not differ significantly between the groups and it decreased in the following 48 h (placebo 71 ± 34 ng/mL and dipeptides 98 ± 30 ng/mL). Mb did not differ between the groups, and the value of 0.05 ≤ *η*^2^ < 0.26 indicated a small effect of dipeptide intake on changes in Mb concentration following intense exercise test. Lactate concentration (*p* < 0.001) increased significantly from 3.3 ± 0.6 to 10.0 ± 2.3 mmol/L in the placebo group and from 2.8 ± 0.7 to 10.4 ± 2.3 mmol/L in the dipeptide group. Lactate level did not differ between the groups.

### 3.4. Oxi-Antioxidant and Inflammatory Variables

The results are presented in Table 3. The post-exercise changes in NO and H_2_O_2_ concentrations proceeded simultaneously and reached their highest values in the 1st min after exercise in the placebo group. The use of dipeptides markedly limited the increase in H_2_O_2_ but did not affect NO during the post-exercise period. This indicates a more significant influence of dipeptides on H_2_O_2_ than on NO generation. Dipeptide intake resulted in a two-fold increase in the NO/H_2_O_2_ ratio in the 1st and 30th min after exercise (Figure 2). The value of effect size 0.026 ≤ *η*^2^ < 0.64 showed a moderate effect of dipeptides on the NO/H_2_O_2_ ratio. 8-Isoprostane, as a marker of oxidative damage, did not change either under the influence of the exercise test or under the influence of dipeptide intake. This could be related to the increase in TAS caused by the release of low-weight antioxidants into the circulation from active skeletal muscles. Indeed, TAS increased in both groups following the exercise test, but it was significantly higher in the group supported by dipeptides. The value of *η*^2^ ≥ 0.64 showed a large effect of dipeptides on TAS before and after the exercise test. Glutathione GSH_t_ behaved similarly in both groups i.e., it significantly decreased in the 1st min after exercise and then increased to reach the pre-exercise level. No differences between dipeptide and placebo groups were recorded. Glutathione GSSG significantly decreased in the 1st and 30th min after exercise; however, the drop was sharper in the dipeptide group. This finally resulted in a significant increase in the thiol redox status in the 1st and 30th min after exercise in both groups (Figure 3). Nevertheless, thiol redox status increased three-fold in the dipeptide group compared to the placebo group, indicating a modifying effect of dipeptides on the blood glutathione profile. Most clinical trials and observational studies have used high-sensitivity CRP as a biochemical marker of inflammation because it is relatively stable and easy to measure. In the present study, CRP concentration was found to fall within normal ranges in all individuals (<5 mg/L). CRP was significantly increased during the post-exercise period compared with the initial analysis. In the dipeptide group, CRP was significantly lower than in the placebo group, which indicates the anti-inflammatory properties of the chicken breast extract.

## 4. Discussion

In sports medicine, the effectiveness of regenerative processes plays a crucial role for athletes’ health and physical performance. Therefore, new food supplements are being sought to modify the cascade of the injury–repair–regeneration of tissues [21]. Our aim was to analyse the effects that dipeptide intake in combination with physical exercise may exert on the oxi-antioxidant response. Chicken meat and its extracts have long been recognized as a source of bioactive dipeptides that can potentially promote tissue regeneration and improve health status [11,21,22,23,24,25,26]. However, the mechanisms underlying the bioactivities of chicken meat could be complicated and may be regulated by the combined actions of many active components [27]. Previous studies have focused on various antiglycating, anti-inflammation, antioxidative, antifatigue and antilipogenic protective properties of chicken-meat-derived dipeptides [16,28,29,30,31,32]. The majority of the available studies have only focused on the antioxidative role of carnosine [33]. For example, diabetic rats which received carnosine showed a 20% reduction in advanced glycation end products, a 24% reduction in serum reactive oxygen and nitrogen species, and a 36% reduction in oxidant-mediated protein damage [33]. Reactive oxygen and nitrogen species production is known to increase dramatically in skeletal muscles, endothelium and erythrocytes following intense exercise [4,34,35]. Furthermore, persistent exercise-induced inflammation in tissues is accompanied by excessive H_2_O_2_ generation and reduced NO bioavailability [4]. In the study, intense exercise significantly elevated concentrations of both reactive molecules: NO and H_2_O_2_. Pre-exercise dipeptide intake reduced post-exercise H_2_O_2_ level but did not affect NO level, which resulted in a higher ratio of NO/H_2_O_2_ compared to the placebo group (Figure 2C,D). A persistently high level of NO is one of the beneficial effects of physical activity. NO produces an anti-inflammatory effect under normal physiological conditions such as physical activity [36]. The studies in human-isolated muscle and myotube culture have demonstrated that changes in NO and H_2_O_2_ within contracting skeletal muscles are key regulators of pre- and posttranslational signalling events leading to the synthesis of cytokines, heat-stress proteins, growth factors and antioxidants [37]. Previously, we demonstrated that changes in the NO/H_2_O_2_ ratio elevated the levels of circulating growth factors especially involved in myogenesis, including PDGF^BB^, IGF-1, HGF, BDNF and VEGF, which are released from leucocytes and muscle cells within a few hours after physical exercise and then secreted from other tissues during the next few days [21].

Chicken extract intake elevated the NO/H_2_O_2_ ratio and the total antioxidant status, and *η*^2^ ≥ 0.64 showed a strong effect of dipeptides on TAS before and after the exercise test. According to Alkhatib et al. [17], the protective antioxidant mechanisms of dipeptides can be attributed to their oxidation reduction effects produced through modulating genes involved glucose and lipid acids metabolism, inflammation, and apoptosis, which was attributed to the increased activity of antioxidant enzymes [38]. Two distinct oxidation reduction pathways have been implicated with the concurrent increase in GSH ability to scavenge RONS, and protection from protein and lipid peroxidation, as well as reduced cell damage [38]. In our study, cellular damage biomarkers, such as Mb and 8-isoprostanes, were all observed to fall within the normal range expected for healthy active individuals [39]. 8-Isoprostanes are prostaglandin-like compounds produced by the RONS-induced oxidation of arachidonic acid. Their detection is supposed to be an important biomarker of oxidative lipid damage. Although an elevated blood 8-isoprostane level was observed after intense exercise [40], in our study, 8-isoprostane did not change either under the influence of exercise test or under the influence of dipeptide intake. Nevertheless, dipeptides showed antioxidant activity by keeping a steady level of GSH_t_. The glutathione antioxidant system has been shown to play an important role in the maintenance of good health and in disease prevention [41]. GSH cannot be administered directly because two pharmaceutical problems have been observed: poor bioavailability with oral administration, and a short half-life (2 min) with intravenous administration [42]. Therefore, various compounds have been used to enhance GSH availability, including diet, nutritional supplementation, and drug administration, all of them with minor to moderate success [43,44,45,46].

The extracellular concentrations of glutathione are usually found to be lower. It may be present in the form of reduced GSH, as well as oxidized disulphide GSSG; however, under normal conditions, almost 99% of cellular glutathione occurs in the form of reduced thiol. Approximately 1–2% of glutathione in cells occur in the oxidized form and this amount increases only under conditions of oxidative stress [47]. In cells, GSH performs a variety of functions that contribute to the maintenance of cellular homeostasis; however, the most important is its intracellular antioxidant activity, where glutathione is the most abundant cellular antioxidant [48]. A limiting factor in cellular GSH synthesis is the availability of cysteine [47]. However, other antioxidants existing as a serum antioxidant status also affect the glutathione pool [43,44,45,49]. According to Elokda and Nielsen [49], thiol redox status is the most sensitive change marker in physical activity. In our study, thiol redox status increased in the 1st and 30th min after exercise in both groups. However, it was three-fold higher in the dipeptide group compared to the placebo group, thereby indicating the modifying effect of dipeptides on the blood glutathione profile (Figure 3C,D).

The changes in RONS and thiol redox status were related to CRP level, which significantly increased during the post-exercise period compared with the initial analysis. Most clinical trials and observational studies have used CRP as a biochemical marker of systemic inflammation that is produced following stimulation by various cytokines that can be the drivers of an acute response to trauma, infection, ischemia and other inflammatory conditions such as physical activity. CRP also upregulates the release of pro-inflammatory cytokines and inhibits the release of NO, which could disrupt tissue regeneration [50]. Nonetheless, our study is the first to demonstrate the anti-inflammatory properties of chicken breast extract. CRP decreased even six-fold in the dipeptide group compared to the placebo group (Table 3). One of recent clinical studies found that hydrolysed chicken extract treatment improved stress-induced cognitive decline and alleviated inflammation in healthy middle-aged humans [51]. Another available study by Ni et al. [52] suggested that chicken meat extract ameliorated neuroinflammation, gliosis, oxidative stress and cognitive decline in naturally aged mice. Rezzani et al. [2], in turn, demonstrated that a carnosine analogue improved oxidative stress, inflammation and cell metabolism, thereby becoming a greatly promising therapeutic molecule. Obviously, the use of chicken breast extract and dipeptides as an immunomodulatory supplement in sport nutrition requires further research, including on the pro- and anti-inflammatory response to intense physical exercise.

## 5. Conclusions

The study demonstrated that high dipeptide intake increases antioxidant status and regulates the oxi-inflammatory response to intense physical exercise. Therefore, the application of dipeptides seems to have favourable therapeutic potential in modulating oxidative stress and inflammation, especially in physically active individuals following a strenuous training schedule.

## 6. Limitations

The limitations of the study include a relatively small number of subjects; nevertheless, this smaller sample proved sufficient to show a protective effect of dipeptides against oxidative stress.

## Figures and Tables

**Figure 1 nutrients-14-02402-f001:**
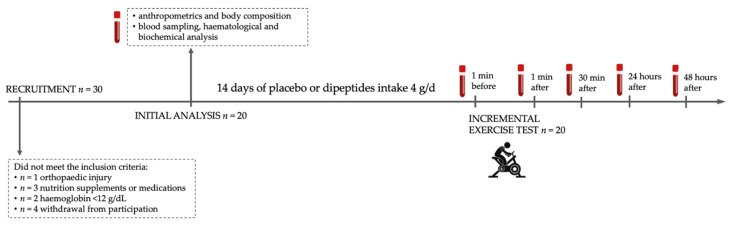
Illustration of the study design.

**Figure 2 nutrients-14-02402-f002:**
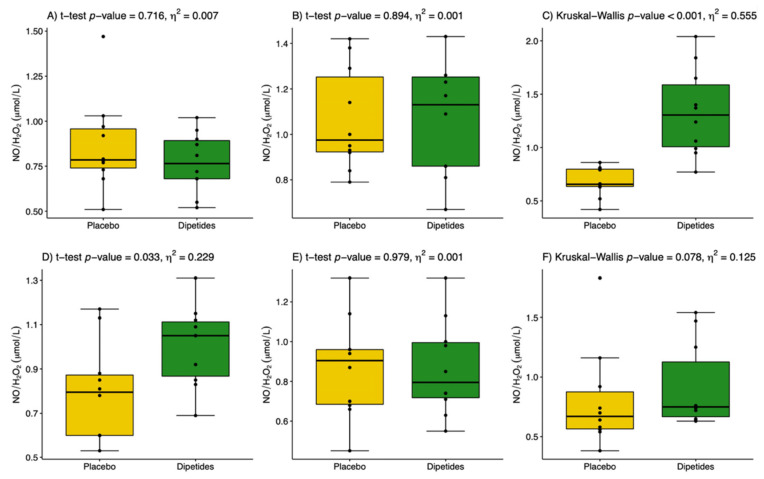
The ratio of nitric oxide to hydrogen peroxide (NO/H_2_O_2_) in placebo (yellow) and dipeptides (green). (**A**) Initial analysis, before placebo or dipeptides intake; (**B**) before incremental exercise test; (**C**) the 1st min after exercise; (**D**) the 30th min after exercise; (**E**) the 24th h after exercise; and (**F**) the 48th h after exercise test. *η*^2^ is a measure of effect size.

**Figure 3 nutrients-14-02402-f003:**
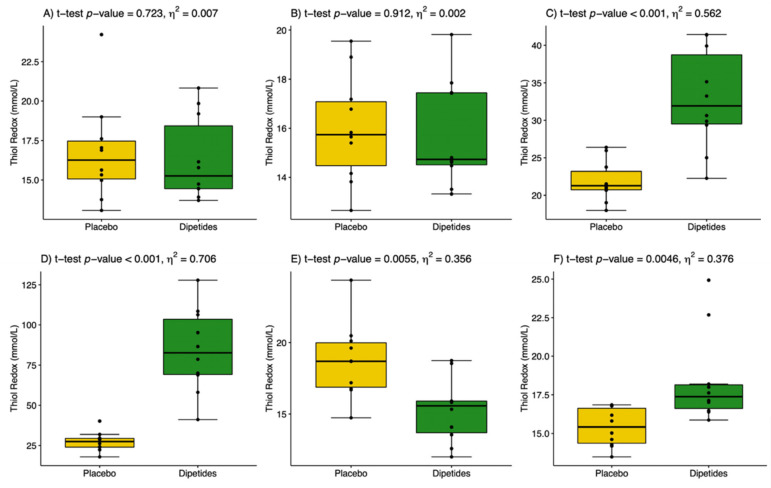
The thiol redox status (the ratio of reduced GSH to oxidized GSSG glutathione) in placebo (yellow) and dipeptides (green). (**A**) Initial analysis, before placebo or dipeptides intake; (**B**) before incremental exercise test; (**C**) the 1st min after exercise; (**D**) the 30th min after exercise; (**E**) the 24th h after exercise; and (**F**) the 48th h after exercise test. *η*^2^ is a measure of effect size.

**Table 1 nutrients-14-02402-t001:** Anthropometrics and body composition (mean ± SD).

	Placebo *n* = 10	Dipeptides *n* = 10	*p*-Value
Age (year)	21.40 ± 2.12	20.25 ± 0.46	0.367
Height (cm)	175.50 ± 12.78	180.40 ± 11.97	0.388
Weight (kg)	71.18 ± 11.71	75.43 ± 18.84	0.482
BMI (kg/m^2^)	23.32 ± 2.68	23.13 ± 3.09	0.887
FM (kg)	14.71 ± 6.39	11.81 ± 8.41	0.307
FFM (kg)	62.27 ± 15.36	64.65 ± 13.12	0.720
VO_2_max (mL/kg/min)	54.79 ± 9.87	55.95 ± 2.48	0.751

Abbreviations: BMI, body mass index; FM, fat mass; FFM, fat-free mass, VO_2_max, maximal oxygen uptake.

**Table 2 nutrients-14-02402-t002:** Haematological variables (mean ± SD).

Variables	Reference Values	Placebo *n* = 10	Dipeptides *n* = 10	*p*-Value
RBC (10^6^/µL)	4.2–6.5	5.10 ±0.06	5.03 ± 0.37	0.577
HB (g/dL)	12.0–18.0	15.01 ± 0.64	14.64 ± 0.55	0.181
HCT%	38.0–54.0	46.05 ± 2.38	45.32 ± 2.51	0.513
MCV fL	80.0–97.0	88.98 ± 2.85	90.20 ± 2.77	0.345
MCH (pg/RBC)	26.0–32.0	29.01 ± 0.91	29.18 ± 1.55	0.768
MCHC (g/dL)	31.0–36.0	32.60 ± 0.32	32.35 ± 1.22	0.546
RDW%	11.5–14.8	13.75 ±0.76	13.73 ± 0.61	0.790
WBC (10^3^/µL)	4.0–10.2	6.06 ± 0.36	5.77 ± 1.06	0.431
PLT (10^3^/µL)	140–420	227 ± 12	226 ± 48	0.352

Abbreviations: SD, standard deviation; RBC, red blood cells; HB, haemoglobin; HCT, haematocrit; MCV, mean cell volume; MCH, mean corpuscular haemoglobin; MCHC, mean corpuscular haemoglobin concentration; RDW, red cell distribution width; WBC, white blood cells; PLT, platelets.

**Table 3 nutrients-14-02402-t003:** Oxi-antioxidant and inflammatory variables (mean ± SD).

			NO (µmol/L)	H_2_O_2_ (µmol/L)	8-Isoprostanes (pg/mL)	TAS (mmol/L)	GSH_t_ (mmol/L)	GSSG (mmol/L)	CRP (mg/L)
Initial level		Placebo	13.99 ± 0.74	16.31 ± 3.53	77.88 ± 10.95	19.48 ± 2.03	1271± 118	65.28 ± 4.72	0.11 ± 0.04
		Dipeptides	12.70 ± 1.57	16.34 ± 3.53	79.67 ± 11.74	19.13 ± 2.21	1191 ± 162	65.30 ± 6.06	0.01 ± 0.04
		*p*-value	0.049	0.983	0.595	0.717	0.238	0.713	0.653
After 14-day placebo or dipeptide intake	Before exercise	Placebo	13.79 ± 0.85	14.91 ± 1.41	72.85 ± 21.05	14.98 ± 0.87	1368 ± 197	76.28 ± 8.46	0.12 ± 0.07
		Dipeptides	15.33 ± 2.60	14.46 ± 0.41	81.46 ± 20.55	22.51 ± 2.15	1474 ± 143	83.61 ± 9.70	0.11 ± 0.04
		*p*-value	0.106	0.382	0.384	*p* < 0.001	0.179	0.090	0.745
	1st min after exercise	Placebo	18.27 ± 1.33	28.79 ± 5.93	93.36 ± 23.56	16.42 ± 1.58	1123 ± 92	47.29 ± 3.22	0.23 ± 0.06
		Dipeptides	19.07 ± 2.32	15.33 ± 4.01	74.42 ± 29.72	20.52 ± 1.74	1091 ± 61	30.35 ± 2.78	0.24 ± 0.07
		*p*-value	0.376	*p* < 0.001	0.161	*p* < 0.001	0.435	*p* < 0.001	0.519
	30th min after exercise	Placebo	17.27 ± 1.51	22.88 ± 5.49	96.34 ± 46.26	16.99 ± 1.93	1335 ± 126	46.47 ± 6.51	0.71 ± 0.16
		Dipeptides	14.03 ± 1.49	13.55 ± 0.93	73.17 ± 15.51	17.00 ± 0.87	1370 ± 163	15.41 ± 2.67	0.36 ± 0.12
		*p*-value	*p* < 0.001	*p* < 0.001	0.910	0.987	0.592	*p* < 0.001	*p* < 0.001
	24th h after exercise	Placebo	14.66 ± 1.79	17.87 ± 2.99	166.82 ± 39.48	12.95 ± 2.05	1222 ±76	58.10 ± 4.31	0.30 ± 0.09
		Dipeptides	15.45 ± 3.00	18.92 ± 5.73	175.43 ± 43.30	18.76 ± 1.63	1223 ± 103	71.60 ± 7.67	0.05 ± 0.008
		*p*-value	0.505	0.629	0.472	*p* < 0.001	0.992	*p* < 0.001	*p* < 0.001
	48th h after exercise	Placebo	13.99 ± 1.23	21.29 ± 2.78	107.11 ± 13.89	11.05 ± 0.53	1345 ± 93	77.42 ± 4.91	0.47 ± 0.12
		Dipeptides	13.87 ± 1.57	17.48 ± 5.41	94.73 ± 35.81	17.88 ± 1.91	1374 ± 83	71.37 ± 3.35	0.07 ± 0.02
		*p*-value	0.862	0.090	0.371	*p* < 0.001	0.479	*p* < 0.001	*p* < 0.001

Abbreviations: NO, nitric oxide; H_2_O_2_, hydrogen peroxide; TAS, total antioxidant status; GSH_t_, total glutathione; GSSG, oxidised glutathione; CRP, C-reactive protein.

## Data Availability

The data used to support the findings of this study are available from the corresponding author upon request.

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
