# Peer review of "Dipeptide Extract Modulates the Oxi-Antioxidant Response to Intense Physical Exercise"

_nutrients, 2022, doi:10.3390/nu14122402_

Round 1

Reviewer 1 Report

The authors in present study propose to evaluate the antioxidant and anti-inflammatory potential of the pre-exercise ingestion of the chicken breast extract as a novel nutritional strategy in physically active individuals. They concluded that high dipeptides intake increases antioxidant status and regulates the oxi-inflammatory response to intensive physical exercise.

Major considerations

The protocol indicates that there is a supplementation with dipeptides or placebo, but there isn’t any information about nutrient intake of participants, this information must be available, because the nutritional information could affect nutritional state of participants.

Minor considerations

Methods: Indicate the nutritionals conditions of participants during bioimpedance analysis  

Results: In my opinion Figure 2, is not necessary, the data could be in a table.  

Conclusions: It is convenient change the affirmation “In this study we demonstrated for the first time that….”  Eliminate “First time”

Author Response

Response to Review 1

We greatly appreciate your time and effort dedicated to providing feedback on our manuscript and we are grateful for the insightful comments on and valuable improvements to our paper. All the suggestions helped us to evaluate our outcomes even more precisely in order to deliver improved, high quality scientific manuscript which we hope will now meet the high standards of Nutrients.

Comments for authors

The authors in present study propose to evaluate the antioxidant and anti-inflammatory potential of the pre-exercise ingestion of the chicken breast extract as a novel nutritional strategy in physically active individuals. They concluded that high dipeptides intake increases antioxidant status and regulates the oxi-inflammatory response to intensive physical exercise.

Major considerations

The protocol indicates that there is a supplementation with dipeptides or placebo, but there isn’t any information about nutrient intake of participants, this information must be available, because the nutritional information could affect nutritional state of participants.

Section 2.1. Participants has been supplemented with the following information concerning the diet analysis:

Throughout the study all subjects lived at the same accommodation and followed the same daily schedule, sleeping time and diet. Dietary intake was evaluated using a food record method (foods and beverages consumed over 24h for 7 consecutive days). The subjects were instructed on how to fill in the food record. The participants self-reported all the consumed foods and fluids and provided information about the times, types, and portion size of the meals and beverages they had consumed. The quantities of all the food/beverage items were reported in household measures or, if available, according to packaging details. The energetic value of food intake, as well as the ingredients of products and meals, were estimated by means of the Dieta 6.0 software (IZZ, Warsaw, Poland). Dieta 6.0 is a valuable application that calculates the intake of different nutrients, with bioavailability taken into account.

Minor considerations

  • Methods: Indicate the nutritionals conditions of participants during bioimpedance analysis.

Section 2.3. Body composition has been supplemented with the following information:

The subjects were fasting during the body composition analysis.

  • Results: In my opinion Figure 2, is not necessary, the data could be in a table. 

Figure 2 has been removed and the results have been described in Section 3.2. Skeletal muscle damage and lactate.

  • Conclusions: It is convenient change the affirmation “In this study we demonstrated for the first time that….”  Eliminate “First time”

The phrase “first time” has been removed.

Reviewer 2 Report

The authors developed an interesting article for the reader of the Journal Nutrients, which added knowledge about on the impact of dipeptide extract on the oxy-antioxidant response to intense physical exercise.

In general, the article is well structured and adequately and succinctly written. However, before its publication, I leave some comments that can be taken into account by the authors:

Abstract:

1- [line 14] The word "Exposure" is in bold. remove bold.

Introduction:

2- In general, the introduction seems to me to be well written and scientifically rigorous. The authors discuss in detail the functions of dipeptide in human organisms, based on scientific evidence. The authors use previously published literature to justify why franco was chosen as the supplement chosen for the study. At the end of the introduction, the objective of the study is presented.

Materials and Methods:

3 - The materials and methods are described in a succinct and complete way, allowing the replication of the study. The characteristics of the sample, all the instruments used, as well as the statistical tests are presented. However, I suggest improving the following points:

4- [Line 73-81] Were calculations performed to estimate the sample size? if yes, they should be described

5- [Line 73-81] What was the randomization method used?

6- [Line 73-81] How was sleep time measured?

7- [Line 73-81] How was the average daily calorie consumption measured?

8- [Line 118-120] How many blood collections were performed? 6? This part of the text seems a little confusing to me.

Results:

9- In my opinion the results are presented correctly. The authors used tables and graphs to present the results obtained, which makes their interpretation more interactive and easier. I liked the fact that the results are subdivided into chapters.

Discussion:

10- In the discussion, the main results obtained and justified with scientific bases verified in previous studies are presented. The results are compared with recent studies. However, in my opinion, the discussion should begin with the presentation of the study objective.

Conclusion:

11- The conclusion is succinct and direct. The authors refer to the main conclusions of the study and reinforce the practical importance of the results obtained for public health, particularly for physically active individuals

I would like to take this opportunity to congratulate the authors for their excellent research work. I wish you all the best and keep up the good work.

Author Response

Response to Review 2

We greatly appreciate your time and effort dedicated to providing feedback on our manuscript and we are grateful for the insightful comments on and valuable improvements to our paper. All the suggestions helped us to evaluate our outcomes even more precisely in order to deliver improved, high quality scientific manuscript which we hope will now meet the high standards of Nutrients.

Comments for authors

The authors developed an interesting article for the reader of the Journal Nutrients, which added knowledge about on the impact of dipeptide extract on the oxy-antioxidant response to intense physical exercise. In general, the article is well structured and adequately and succinctly written. However, before its publication, I leave some comments that can be taken into account by the authors:

Abstract:

1- [line 14] The word "Exposure" is in bold. remove bold.

Abstract has been re-inserted into the manuscript to eliminate the editorial errors.

Introduction:

2- In general, the introduction seems to me to be well written and scientifically rigorous. The authors discuss in detail the functions of dipeptide in human organisms, based on scientific evidence. The authors use previously published literature to justify why franco was chosen as the supplement chosen for the study. At the end of the introduction, the objective of the study is presented.

Thank you very much for your comment.

Materials and Methods:

3 - The materials and methods are described in a succinct and complete way, allowing the replication of the study. The characteristics of the sample, all the instruments used, as well as the statistical tests are presented. However, I suggest improving the following points:

4- [Line 73-81] Were calculations performed to estimate the sample size? if yes, they should be described.

Yes, we calculated the sample size. Section 2.9. Statistical analysis has been completed accordingly: The ideal sample size for a 30-person group was n=28 whereas for a 20-person it was set at n=20 to demonstrate significance differences (confidence level 95% and margin error 5%).

5- [Line 73-81] What was the randomization method used?

Random assignment refers to the use of chance procedures in medical experiments to ensure that each participant has the same opportunity to be assigned to any given group. Study participants are randomly assigned to different groups, such as the experimental group or treatment group.

We based on a single sequence of random assignments which is known as simple randomization described by Kang M. et al. J Athl Train 2008. The method included using a shuffled deck of cards (placebo or dipeptide) with numbers from 1 to 20. Then, the random numbers/cards were found in the random number table. The section 2.1. Participants has been supplemented with the randomization technique used.

6- [Line 73-81] How was sleep time measured?

Seven hours of sleep were recommended for subjects according to Connor TJ. Don’t stress out your immune system; just relax. Bran, Behaviour, and Immunity 2008, to reduce the generation of inflammatory mediators.

7- [Line 73-81] How was the average daily calorie consumption measured?

The mean daily consumption of energy did not exceed 2800 kcal.

8- [Line 118-120] How many blood collections were performed? 6? This part of the text seems a little confusing to me.

According to Figure 1, the blood samples were collected 6 times. The information has been clarified in Section 2.5. Blood sampling.

Results:

9- In my opinion the results are presented correctly. The authors used tables and graphs to present the results obtained, which makes their interpretation more interactive and easier. I liked the fact that the results are subdivided into chapters.

Thank you very much for your comment.

Discussion:

10- In the discussion, the main results obtained and justified with scientific bases verified in previous studies are presented. The results are compared with recent studies. However, in my opinion, the discussion should begin with the presentation of the study objective.

The section 4.Discussion has been revised.

Conclusion:

11- The conclusion is succinct and direct. The authors refer to the main conclusions of the study and reinforce the practical importance of the results obtained for public health, particularly for physically active individuals.

Thank you very much for your comment.

I would like to take this opportunity to congratulate the authors for their excellent research work. I wish you all the best and keep up the good work.

Round 2

Reviewer 1 Report

Authors have improved the article in base of all coments.

Author Response

Response to Reviewer 1

We greatly appreciate your time and effort dedicated to providing feedback on our manuscript and we are grateful for the insightful comments on and valuable improvements to our paper. All the suggestions helped us to evaluate our outcomes even more precisely in order to deliver improved, high quality scientific manuscript which we hope will now meet the high standards of Nutrients.

Comments for authors

The authors in present study propose to evaluate the antioxidant and anti-inflammatory potential of the pre-exercise ingestion of the chicken breast extract as a novel nutritional strategy in physically active individuals. They concluded that high dipeptides intake increases antioxidant status and regulates the oxi-inflammatory response to intensive physical exercise.

Major considerations

The protocol indicates that there is a supplementation with dipeptides or placebo, but there isn’t any information about nutrient intake of participants, this information must be available, because the nutritional information could affect nutritional state of participants.

The sections Material and methods (2.2.Diet analysis) and Results (3.1.Diet analysis) have been supplemented with the following information concerning the diet analysis:

The diet was evaluated using a food record method (foods and beverages consumed over 24 h for 7 consecutive days). The subjects were instructed on how to fill in the food record. The participants self-reported all the consumed foods and fluids and provided information about the times, types, and portion size of the meals and beverages they had consumed. The quantities of all the food/beverage items were reported in household measures or, if available, according to packaging details. The energetic value of food intake, as well as the ingredients of products and meals, were estimated by means of the Dieta 6.0 software (IZZ, Warsaw, Poland). Dieta 6.0 is a valuable application that calculates the intake of different nutrients, with bioavailability taken into account. To evaluate compliance with the recommended dietary intake, the supply of different nutrients was categorized as intake in accordance with the nutritional standards for Polish population [Jarosz, M.; Rychlik, E.; Stoś, K.; Charzewska, J. Normy żywienia dla populacji Polski i ich zastosowanie; Narodowy Instytut Zdrowia Publicznego: Warszawa, 2020; ISBN 978-83-65870-28-5].

According to the standard of nutrition for the Polish adult population [Jarosz, M.; Rychlik, E.; StoÅ›, K.; Charzewska, J. Normy żywienia dla populacji Polski i ich zastosowanie; Narodowy Instytut Zdrowia Publicznego: Warszawa, 2020; ISBN 978-83-65870-28-5], our results did not demonstrate differences in intake of the major ingredients that might influnce the oxi-antioxidant response. The mean daily consumption of energy, protein, carbohydrates and fat was 32.5 ± 6.5 kcal/kg body mass/day, 1.5 ± 0.3 g/kg body mass/day, 3.7 ± 0.8 g/kg body mass/day, 1.1 ± 0.3 g/kg body weight, respectively.

Minor considerations

  • Methods: Indicate the nutritionals conditions of participants during bioimpedance analysis.

Section 2.3. Body composition has been supplemented with the following information:

The subjects were fasting during the body composition analysis.

  • Results: In my opinion Figure 2, is not necessary, the data could be in a table. 

Figure 2 has been removed and the results have been described in Section 3.2. Skeletal muscle damage and lactate.

  • Conclusions: It is convenient change the affirmation “In this study we demonstrated for the first time that….”  Eliminate “First time”

The phrase “first time” has been removed.
